# Smoking, alcohol consumption, and 24 gastrointestinal diseases: Mendelian randomization analysis

**Shuai Yuan[1,2†], Jie Chen[1,3†], Xixian Ruan[3†], Yuhao Sun[1], Ke Zhang[4,5], Xiaoyan Wang[3*], Xue Li[1,6*], Dipender Gill[7], Stephen Burgess[8,9], Edward Giovannucci[10,11], Susanna C Larsson[2,12]**

[1]School of Public Health and The Second Affiliated Hospital, Zhejiang University School of Medicine, Zhejiang, China; [2]Unit of Cardiovascular and Nutritional Epidemiology, Institute of Environmental Medicine, Karolinska Institutet, Stockholm, Sweden; [3]Department of Gastroenterology, The Third Xiangya Hospital, Central South University, Changsha, China; [4]Key Laboratory of Growth Regulation and Translational Research of Zhejiang Province, School of Life Sciences, Westlake University, Hangzhou, China; [5]Westlake Intelligent Biomarker Discovery Lab, Westlake Laboratory of Life Sciences and Biomedicine, Hangzhou, China; [6]Centre for Global Health Research, Usher Institute, University of Edinburgh, Edinburgh, United Kingdom; [7]Department of Epidemiology and Biostatistics, School of Public Health, Imperial College London, London, United Kingdom; [8]MRC Biostatistics Unit, University of Cambridge, Cambridge, United Kingdom; [9]Department of Public Health and Primary Care, University of Cambridge, Cambridge, United Kingdom; [10]Department of Epidemiology, Harvard T.H. Chan School of Public Health, Boston, United States; [11]Department of Nutrition, Harvard T.H. Chan School of Public Health, Boston, United States; [12]Unit of Medical Epidemiology, Department of Surgical Sciences, Uppsala University, Uppsala, Sweden

*For correspondence:
xue.li@ed.ac.uk (XL);
wangxiaoyan@csu.edu.cn (XW)

†These authors contributed equally to this work

Competing interest: The authors declare that no competing interests exist.

## Abstract

**Background:** Whether the positive associations of smoking and alcohol consumption with gastrointestinal diseases are causal is uncertain. We conducted this Mendelian randomization (MR) to comprehensively examine associations of smoking and alcohol consumption with common gastrointestinal diseases.

**Methods:** Genetic variants associated with smoking initiation and alcohol consumption at the genome-wide significance level were selected as instrumental variables. Genetic associations with 24 gastrointestinal diseases were obtained from the UK Biobank, FinnGen study, and other large consortia. Univariable and multivariable MR analyses were conducted to estimate the overall and independent MR associations after mutual adjustment for genetic liability to smoking and alcohol consumption.

**Results:** Genetic predisposition to smoking initiation was associated with increased risk of 20 of 24 gastrointestinal diseases, including 7 upper gastrointestinal diseases (gastroesophageal reflux, esophageal cancer, gastric ulcer, duodenal ulcer, acute gastritis, chronic gastritis, and gastric cancer), 4 lower gastrointestinal diseases (irritable bowel syndrome, diverticular disease, Crohn's disease, and ulcerative colitis), 8 hepatobiliary and pancreatic diseases (non-alcoholic fatty liver disease, alcoholic liver disease, cirrhosis, liver cancer, cholecystitis, cholelithiasis, and acute and chronic pancreatitis), and acute appendicitis. Fifteen out of 20 associations persisted after adjusting for genetically

predicted alcohol consumption. Genetically predicted higher alcohol consumption was associated with increased risk of duodenal ulcer, alcoholic liver disease, cirrhosis, and chronic pancreatitis; however, the association for duodenal ulcer did not remain statistically significant after adjustment for genetic predisposition to smoking initiation.

**Conclusions:** This study provides MR evidence supporting causal associations of smoking with a broad range of gastrointestinal diseases, whereas alcohol consumption was associated with only a few gastrointestinal diseases.

**Funding:** The Natural Science Fund for Distinguished Young Scholars of Zhejiang Province; National Natural Science Foundation of China; Key Project of Research and Development Plan of Hunan Province; the Swedish Heart Lung Foundation; the Swedish Research Council; the Swedish Cancer Society.

## Editor's evaluation

This is a valuable article that is methodologically convincing and provides evidence, through Mendelian Randomisation, that genetic predisposition to smoking and alcohol consumption influences the risk to develop different gastrointestinal diseases. The findings largely corroborate the findings from observational studies, especially for the effects of smoking. The major strength of the paper is the use of the largest possible genetic datasets for both the exposures and outcomes, which makes the findings more robust.

## Introduction

Tobacco smoking and alcohol consumption are leading causes of the global burden of disease and are major contributors to premature mortality (*GBD 2016 Alcohol Collaborators, 2018*; *GBD 2016 Alcohol Collaborators, 2020*). Gastrointestinal diseases account for considerable health care use and expenditures, and a holistic approach to lifestyle interventions may result in more health gains and less economic burdens (*Peery et al., 2022*). Population-based studies have identified tobacco smoking as a risk factor for several gastrointestinal diseases, including gastroesophageal reflux disease (*Eusebi et al., 2018*), esophageal cancer (*Fund WCR and Research AlfC, 2007*), Crohn's disease (*Piovani et al., 2019*), liver cancer (*McGee et al., 2019*), and pancreatitis (*Yadav and Whitcomb, 2010*). Evidence on the association between tobacco smoking and risk of other gastrointestinal diseases is limited and inconsistent. Like smoking, heavy alcohol consumption has been associated with increased risk of several gastrointestinal outcomes, including gastritis (*Bujanda, 2000*), gastric cancer (*Laszkowska et al., 2021*), colorectal cancer (*McNabb et al., 2020*), cirrhosis (*Simpson et al., 2019*), liver cancer (*McGee et al., 2019*), and pancreatitis (*Yadav and Whitcomb, 2010*). However, whether these associations are all causal remains unestablished, since most of the evidence was obtained from observational studies where the results may be biased by reverse causality and confounding. Of note, even though reverse causality may not be an issue in the studies for any of studied gastroenterological outcomes, it might exist for certain gastroenterological diseases causing pain, which smoker patients may try to increase smoking dose to mitigate via an intake of higher levels of nicotine. In addition, as smoking and alcohol consumption are phenotypically and genetically correlated (*Roberts et al., 2020*; *Liu et al., 2019*), the independent impacts of smoking and alcohol consumption on gastrointestinal diseases are unclear. Establishing the causal association of tobacco smoking and alcohol consumption with gastrointestinal diseases is crucial, as this provides further evidence for subsequent recommending public policies and clinical interventions.

Mendelian randomization (MR) is an epidemiological approach that utilizes genetic variants as an instrument to strengthen the causal inference in an exposure-outcome association (*Davey Smith and Hemani, 2014*). MR is by nature not prone to confounding since genetic variants are randomly assorted at conception and thus unrelated to environmental and self-adopted factors that usually act as confounders. Additionally, this method can minimize reverse causality since fixed alleles are unaffected by the onset and progression of disease. Previous MR studies have examined the associations of smoking and alcohol consumption with several gastrointestinal diseases (*Yuan and Larsson, 2022a*; *Larsson et al., 2020*; *Yuan and Larsson, 2022b*; *Yuan et al., 2022c*; *Chen et al., 2022*; *Yuan et al., 2021*). Nevertheless, whether smoking and alcohol consumption exert influence on a wide range of

**eLife digest** People who smoke cigarettes or drink large amounts of alcohol are more likely to develop disorders with their digestive system. But it is difficult to prove that heavy drinking or smoking is the primary cause of these gastrointestinal diseases.

For example, it is possible that having a digestive disorder makes people more likely to take up these habits to reduce pain or discomfort caused by the illness (an effect known as reverse causation). The association may also be the result of confounding factors, such as age or diet, which contribute to digestive problems as well as the health outcomes of smoking and drinking. Additionally, many people who smoke also drink alcohol and vice versa, making it challenging to determine if one or both behaviors contribute to the disease.

One solution is to employ Mendelian randomization which uses genetics to determine if two variables are linked. Using this statistical approach, Yuan, Chen, Ruan et al. investigated if people who display genetic variants that predispose someone to becoming a smoker or drinker are at greater risk of developing certain digestive disorders. This reduces the possibility of confounding and reverse causation, as any association between genetic variants will have been present since birth, and will have not been impacted by external factors.

Yuan, Chen, Ruan et al. used data from two studies that had collected the genetic and health information of thousands of people living in the United Kingdom or Finland. The analyses revealed that genetic variants associated with cigarette smoking increase the risk of 20 of the 24 gastrointestinal diseases investigated. This risk persisted for most of the disorders, even after adjusting for genes linked with alcohol consumption.

Further analysis showed that genetic variants linked to heavy drinking increase the risk of duodenal ulcer, alcoholic liver disease, cirrhosis, and chronic pancreatitis. However, accounting for smoking-linked genes eliminated the relationship with duodenal ulcer.

These findings suggest that smoking has detrimental effects on gastrointestinal health. Reducing the number of people who start smoking or encouraging smokers to quit may help prevent digestive diseases. Even though there were fewer associations between heavy alcohol consumption and gastrointestinal illness, further studies are needed to investigate this relationship in more depth.

gastrointestinal outcomes has not been investigated in a comprehensive way. A thorough investigation on the gastrointestinal consequences of smoking and alcohol drinking is of great importance to develop non-pharmacological interventions on gastrointestinal diseases. Here, we conducted an MR investigation of the associations of smoking and alcohol consumption with the risk of common gastrointestinal diseases to fill in above knowledge gaps.

## Materials and methods

*Figure 1* shows the study design overview. The study was based on publicly available genome-wide association studies (GWAS), and the detailed information on used studies was presented in *Supplementary file 1A*. The genetic associations were estimated using data from the UK Biobank study (*Sudlow et al., 2015*), the FinnGen study (*Kurki et al., 2022*; https://www.finngen.fi/en), and several large consortia. The summary effect estimates were combined using meta-analysis for each gastrointestinal disease from different data resources. Included studies had been approved by corresponding institutional review boards and ethical committees, and consent forms had been signed by all participants.

### Instrumental variable selection

A total of 378 and 99 single nucleotide polymorphisms (SNPs) associated with smoking initiation (a binary phenotype indicating whether an individual had ever being a regular smoker, 1,232,091 individuals of European descent) and alcohol consumption (log-transformed drinks per week, 941,280 individuals of European descent) at the genome-wide significance threshold ($p<5 \times 10^{-8}$) were identified by the GWAS and Sequencing Consortium of Alcohol and Nicotine use (GSCAN) study (*Liu et al., 2019*). These SNPs explained approximately 2.3 and 0.3% of the variation in smoking initiation

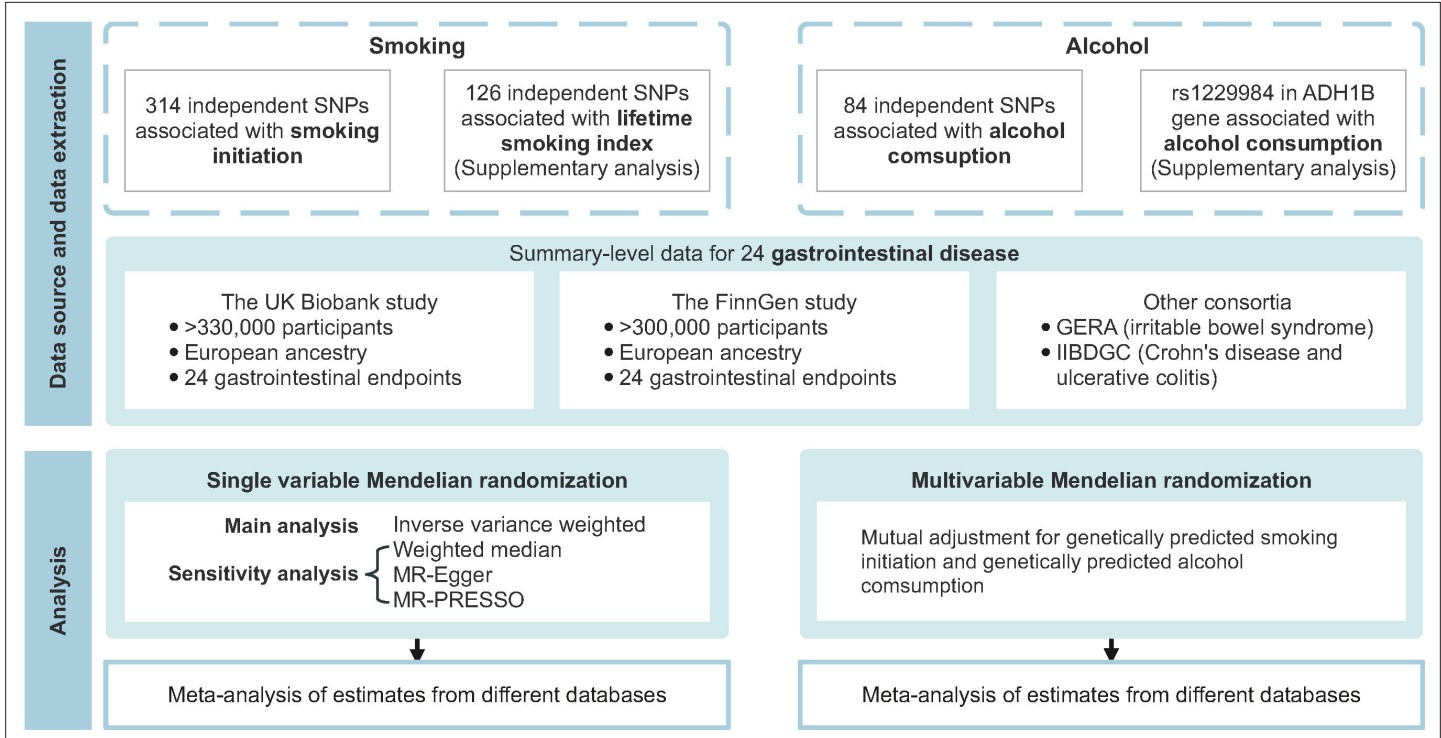

**Figure 1.** Overview of the present study design. GERA, Genetic Epidemiology Research on Aging; IIBDGC, the International Inflammatory Bowel Disease Genetics Consortium; MR, Mendelian randomization; MR-PRESSO, Mendelian randomization pleiotropy residual sum and outlier; SNP, single nucleotide polymorphism.

and alcohol consumption, respectively (*Liu et al., 2019*). SNPs in linkage disequilibrium (defined as $r^2$ >0.01 or clump distance <10,000 kb) and with the weaker associations with the exposure were removed, leaving 314 independent SNPs as instrumental variables for smoking initiation and 84 for alcohol consumption. Smoking initiation and alcohol consumption shared two index genetic variants, which were rs1713676 and rs11692435. Considering partial sample overlap (around 30%) between the GSCAN study with full data and the UK Biobank study (*Liu et al., 2019*), we performed sensitivity analyses for smoking initiation and alcohol consumption using summary statistics data from the analysis excluding the UK Biobank and 23andMe. For a supplementary analysis of smoking behavior, we used 126 SNPs associated with a lifetime smoking index that considered smoking duration, heaviness, and cessation (*Wootton et al., 2020*). The set of genetic instruments captured around 0.36% of the variance in lifetime smoking (*Wootton et al., 2020*). We also conducted a sensitivity analysis using rs1229984 in *ADH1B* gene that encodes alcohol dehydrogenase 1B enzyme as the genetic instrument for alcohol consumption to minimize pleiotropy. Detailed information on used SNPs is presented in *Supplementary file 1B*.

## Gastrointestinal disease data sources

Genetic associations with 24 gastrointestinal diseases were obtained from the UK Biobank study (*Sudlow et al., 2015*), the FinnGen study (*Kurki et al., 2022*), and two large consortia, including the International Inflammatory Bowel Disease Genetics Consortium (IIBDGC) (*Liu et al., 2015*) and Genetic Epidemiology Research on Aging (GERA) (*Guindo-Martínez et al., 2021*). Included outcomes were classified into four major categories according to the disease onset site: (1) upper gastrointestinal diseases (gastroesophageal reflux disease, esophageal cancer, gastric ulcer, acute gastritis, chronic gastritis, and gastric cancer); (2) lower gastrointestinal diseases (irritable bowel disease, celiac disease, diverticular disease, Crohn's disease, ulcerative colitis, and colorectal cancer); (3) hepatobiliary and pancreatic diseases (non-alcoholic fatty liver disease, alcoholic liver disease, cirrhosis, liver cancer, cholangitis, cholecystitis, cholelithiasis, acute pancreatitis, chronic pancreatitis, and pancreatic cancer); and (4) other (acute appendicitis).

The UK Biobank study is a large multicenter cohort study of 500,000 participants recruited in the United Kingdom between 2006 and 2010 (*Sudlow et al., 2015*). We used the summary statistics of European ancestry from GWAS conducted by Lee lab, where the gastrointestinal outcomes were defined by codes of the International Classification of Diseases 9th Revision (ICD-9) and ICD-10 (*Zhou et al., 2020*). Genetic associations were estimated by logistic regression with adjustment for sex, birth year, and the first four genetic principal components. For the FinnGen study, we used summary-level data on the genetic associations with gastrointestinal diseases from the last publicly available R7 data release (*Kurki et al., 2022*). The FinnGen study is a nationwide genetic study where genetic and electronic health record data were collected. The gastrointestinal diseases were ascertained by the codes of the ICD-8, ICD-9, and ICD-10. Genome-wide association analyses were adjusted for sex, age, genetic components, and genotyping batch. Summary-level genetic data on Crohn's disease (5956 cases and 14,927 controls) and ulcerative colitis (6968 cases and 20,464 controls) were additionally obtained from the IIBDGC (*Liu et al., 2015*), and data on irritable bowel syndrome (3117 cases and 53,520 controls) were obtained from the GERA (*Guindo-Martínez et al., 2021*). Detailed diagnostic codes are listed in *Supplementary file 1C*.

## Statistical analysis

Data were harmonized to omit ambiguous SNPs with non-concordant alleles and palindromic SNPs with ambiguous minor allele frequency (>0.42 and <0.58) were removed from the analysis. The primary MR analyses were performed by the multiplicative random-effects inverse-variance weighted (IVW) method, which provides the most precise estimates though assuming that all SNPs are valid instruments. The analysis of rs1229984 for alcohol consumption was conducted by the Wald method. Estimates for each association from different sources were combined using fixed-effects meta-analysis, and the heterogeneity of the associations from different data sources was evaluated by the $I^2$ statistic. Heterogeneity among SNPs' estimates in each association was assessed by Cochran's Q value. Multivariable MR analyses were conducted to mutually adjust for smoking initiation and alcohol consumption. To detect potential unbalanced pleiotropy (horizontal pleiotropy) and examine the consistency of the associations, three sensitivity analyses including the weighted median (*Yavorska and Burgess, 2017*), MR-Egger (*Burgess and Thompson, 2017*), and MR pleiotropy residual sum and outlier (MR-PRESSO) (*Verbanck et al., 2018*) analyses were performed. The weighted median method can provide consistent estimates when more than 50% of the weight comes from valid instrument variants (*Yavorska and Burgess, 2017*). The MR-Egger intercept test can detect unmeasured pleiotropy, and MR-Egger regression can generate estimates after accounting for horizontal pleiotropy albeit with less precision (*Burgess and Thompson, 2017*). The MR-PRESSO method can identify SNP outliers and provide results identical to that from IVW after removal of outliers (*Verbanck et al., 2018*). The F-statistic was estimated to quantify instrument strength, and an F-statistic >10 suggested a sufficiently strong instrument. Power analysis was performed using an online tool (*Brion et al., 2013*). The Benjamini-Hochberg correction that controls the false discovery rate was applied to correct for multiple testing. The association with a nominal p-value <0.05 but Benjamini-Hochberg adjusted p-value >0.05 was regarded suggestive, and the association with a Benjamini-Hochberg adjusted p-value <0.05 was deemed significant. All analyses were two-sided and performed using the TwoSampleMR (*Hemani et al., 2018*), MendelianRandomization (*Yavorska and Burgess, 2017*), and MRPRESSO R packages (*Verbanck et al., 2018*) in R software 4.1.2.

## Results

The F-statistic for each genetic variant was above 10, suggesting a good strength of used genetic instruments (*Supplementary file 1B*). Most associations were well powered (*Supplementary file 1D*). For smoking initiation, there was 80% power to detect the smallest odds ratio (OR) ranging from 1.08 to 1.40 for included outcomes. Although power was lower for alcohol consumption, it was adequate to detect a moderate effect size for most common gastrointestinal diseases.

### Smoking and gastrointestinal diseases

Genetic predisposition to smoking initiation was associated with 20 of the 24 studied gastrointestinal diseases, and all these associations remained after multiple comparison correction (*Table 1* and

*Supplementary file 1E*). In detail, genetic liability to smoking initiation was positively associated with seven upper gastrointestinal diseases: gastroesophageal reflux (OR, 1.28; 95% confidence interval [CI], 1.20–1.37; p=4.09 × 10$^{-14}$), esophageal cancer (OR, 1.67; 95% CI, 1.24–2.25; p=6.84 × 10$^{-4}$), gastric ulcer (OR, 1.54; 95% CI, 1.37–1.72; p=3.83 × 10$^{-14}$), duodenal ulcer (OR, 1.53; 95% CI, 1.34–1.75; p=8.47 × 10$^{-10}$), acute gastritis (OR, 1.29; 95% CI, 1.09–1.53; p=0.003), chronic gastritis (OR, 1.33; 95% CI, 1.18–1.49; p=1.55 × 10$^{-6}$), and gastric cancer (OR, 1.42; 95% CI, 1.13–1.79; p=0.003); genetic liability to smoking initiation was positively associated with four lower gastrointestinal diseases: irritable bowel syndrome (OR, 1.22; 95% CI, 1.12–1.32; p=3.50 × 10$^{-6}$), diverticular disease (OR, 1.25; 95% CI, 1.18–1.33; p=5.23 × 10$^{-14}$), Crohn's disease (OR, 1.25; 95% CI, 1.11–1.40; p=3.03 × 10$^{-4}$), and ulcerative colitis (OR, 1.15; 95% CI, 1.04–1.26; p=0.004); genetic liability to smoking initiation was positively associated with eight hepatobiliary and pancreatic diseases: non-alcoholic fatty liver disease (OR, 1.49; 95% CI, 1.26–1.76; p=3.82 × 10$^{-6}$), alcoholic liver disease (OR, 1.99; 95% CI, 1.65–2.41; p=1.49 × 10$^{-12}$), cirrhosis (OR, 1.68; 95% CI, 1.40–2.02; p=3.39 × 10$^{-8}$), liver cancer (OR, 1.57; 95% CI, 1.13–2.17; p=0.007), cholecystitis (OR, 1.47; 95% CI, 1.29–1.68; p=4.71 × 10$^{-9}$), cholelithiasis (OR, 1.20; 95% CI, 1.13–1.27; p=5.75 × 10$^{-9}$), acute pancreatitis (OR, 1.39; 95% CI, 1.23–1.56; p=6.71 × 10$^{-8}$), and chronic pancreatitis (OR, 1.38; 95% CI, 1.17–1.64; p=1.79 × 10$^{-4}$); genetic liability to smoking initiation was positively associated with acute appendicitis (OR, 1.15; 95% CI, 1.08–1.23; p=1.27 × 10$^{-5}$). Results were consistent in sensitivity analyses. An indication of horizontal pleiotropy was observed in the analysis of esophageal cancer in the FinnGen study (p for MR-Egger intercept <0.05, *Supplementary file 1F*). Although MR-PRESSO detected one to three outliers, the associations persisted and remained significant after removal of these out-lying SNPs (*Supplementary file 1F*). When using the genetic variants for smoking initiation based on data without the UK Biobank and 23andMe studies, the associations attenuated slightly albeit remained significant after multiple comparisons (*Supplementary file 1L* and *Supplementary file 1G*). All associations were replicated in the supplementary analysis of the lifetime smoking index (*Supplementary file 1G*). After correcting for multiple testing, genetically predicted lifetime smoking index was significantly associated with 17 of 24 gastrointestinal diseases, where the patterns of associations were generally similar to the analysis for smoking initiation (*Supplementary file 1M* and *Supplementary file 1G*). In distinction to the analysis of smoking initiation, genetically predicted lifetime smoking index was not significantly associated with acute gastritis, gastric cancer, Crohn's disease, and ulcerative colitis, whereas genetically predicted lifetime smoking index was significantly associated with pancreatic cancer (OR, 2.09; 95% CI, 1.30–3.36).

In multivariable MR analysis adjusted for genetically predicted alcohol consumption, the associations between genetically predicted smoking initiation and gastrointestinal diseases were consistent with that from univariable MR analysis (*Table 1* and *Supplementary file 1H*). However, the associations became stronger with wider CIs, in particular the associations for gastrointestinal reflux, esophageal cancer, gastric ulcer, irritable bowel syndrome, diverticular disease, non-alcoholic fatty liver disease, alcoholic liver disease, and cholecystitis (*Table 1*). In addition, the association for pancreatic cancer became suggestive significant from null.

## Alcohol consumption and gastrointestinal diseases

Genetically predicted alcohol consumption was nominally positively associated with esophageal cancer (OR, 2.86; 95% CI, 1.18–6.91; p=0.020), duodenal ulcer (OR, 1.92; 95% CI, 1.23–3.00; p=0.004), alcoholic liver disease (OR, 14.35; 95% CI, 7.69–26.81; p=6.32 × 10$^{-17}$), cirrhosis (OR, 2.96; 95% CI, 1.50–5.85; p=0.002), and chronic pancreatitis (OR, 2.96; 95% CI, 1.80–4.89; p=2.13 × 10$^{-5}$), and nominally inversely associated with irritable bowel disease (OR, 0.73; 95% CI 0.57–0.93; p=0.012) (*Table 2*). After Benjamini-Hochberg correction, the associations for duodenal ulcer, alcoholic liver disease, cirrhosis, and chronic pancreatitis remained (*Supplementary file 1E*). Results were consistent in sensitivity analyses, and no horizontal pleiotropy was detected (*Supplementary file 1I*). One outlier was detected in the analysis of duodenal ulcer in the FinnGen study, and the association slightly changed after removal of this outlier (*Supplementary file 1I*). Results were consistent in the sensitivity analysis, where the genetic associations with alcohol consumption were obtained from the genome-wide association analysis excluding the UK Biobank and 23andMe studies (*Supplementary file 1N* and *Supplementary file 1G*). The associations were directionally consistent albeit with wider CIs in the analysis, where alcohol consumption was instrumented by rs1229984 (*Supplementary file 1J*). The associations for alcoholic liver disease, cirrhosis, and chronic pancreatitis persisted after adjustment

**Table 1.** Associations of genetic predisposition to smoking initiation with 24 gastrointestinal diseases in univariable and multivariable Mendelian randomization analyses.

| Disease | | Total cases | Total controls | UVMR OR (95% CI) | p Value | I$^2$ (95% CI) | MVMR adjusted for alcohol consumption OR (95% CI) | p Value |
|---|---|---|---|---|---|---|---|---|
| Upper gastrointestinal diseases | Gastroesophageal reflux | 34,135 | 634,629 | 1.28 (1.20, 1.37) | $4.09 \times 10^{-14*}$ | 46.24 | 1.65 (1.35, 2.02) | $1.38 \times 10^{-6*}$ |
| | Esophageal cancer | 1130 | 702,116 | 1.67 (1.24, 2.25) | $6.84 \times 10^{-4*}$ | 22.68 | 4.78 (2.10, 10.90) | $1.97 \times 10^{-4*}$ |
| | Gastric ulcer | 8651 | 666,879 | 1.54 (1.37, 1.72) | $3.83 \times 10^{-14*}$ | 44.96 | 1.95 (1.40, 2.71) | $7.31 \times 10^{-5*}$ |
| | Duodenal ulcer | 5713 | 666,879 | 1.53 (1.34, 1.75) | $8.47 \times 10^{-10*}$ | 0.00 | 1.64 (1.07, 2.52) | 0.024 |
| | Acute gastritis | 3048 | 643,478 | 1.29 (1.09, 1.53) | 0.003* | 0.00 | 1.54 (0.91, 2.62) | 0.106 |
| | Chronic gastritis | 7975 | 643,478 | 1.33 (1.18, 1.49) | $1.55 \times 10^{-6*}$ | 77.04 | 1.33 (0.96, 1.86) | 0.091 |
| | Gastric cancer | 1608 | 701,472 | 1.42 (1.13, 1.79) | 0.003* | 0.00 | 2.29 (1.14, 4.59) | 0.020 |
| Lower gastrointestinal diseases | Irritable bowel disease | 15,718 | 641,489 | 1.22 (1.12, 1.32) | $3.50 \times 10^{-6*}$ | 11.84 | 1.43 (1.10, 1.85) | 0.008* |
| | Celiac disease | 4808 | 631,700 | 0.82 (0.66, 1.02) | 0.071 | 0.00 | 0.87 (0.53, 1.43) | 0.590 |
| | Diverticular disease | 50,065 | 587,969 | 1.25 (1.18, 1.33) | $5.23 \times 10^{-14*}$ | 67.29 | 1.56 (1.30, 1.87) | $1.41 \times 10^{-6*}$ |
| | Crohn's disease | 10,846 | 645,718 | 1.25 (1.11, 1.40) | $3.03 \times 10^{-4*}$ | 0.00 | 1.48 (1.01, 2.16) | 0.042 |
| | Ulcerative colitis | 16,770 | 651,255 | 1.15 (1.04, 1.26) | 0.004* | 0.00 | 0.94 (0.71, 1.25) | 0.677 |
| | Colorectal cancer | 9519 | 686,953 | 1.03 (0.92, 1.14) | 0.632 | 29.94 | 1.03 (0.76, 1.39) | 0.841 |

*Table 1 continued on next page*

*Table 1 continued*

| Disease | | Total cases | Total controls | UVMR OR (95% CI) | p Value | I² (95% CI) | MVMR adjusted for alcohol consumption OR (95% CI) | p Value |
|---|---|---|---|---|---|---|---|---|
| Hepatobiliary and pancreatic diseases | Non-alcoholic fatty liver disease | 3242 | 707,631 | 1.49 (1.26, 1.76) | $3.82 \times 10^{-6}$* | 0.00 | 2.11 (1.15, 3.88) | 0.016* |
| | Alcoholic liver disease | 2955 | 680,369 | 1.99 (1.65, 2.41) | $1.49 \times 10^{-12}$* | 92.68 | 2.26 (1.26, 4.03) | 0.006 |
| | Cirrhosis | 5904 | 706,200 | 1.68 (1.40, 2.02) | $3.39 \times 10^{-8}$* | 0.00 | 1.92 (1.06, 3.47) | 0.032 |
| | Liver cancer | 714 | 702,008 | 1.57 (1.13, 2.17) | 0.007* | 0.00 | 1.96 (0.73, 5.25) | 0.183 |
| | Cholangitis | 1708 | 664,749 | 1.02 (0.80, 1.29) | 0.892 | 0.00 | 1.31 (0.61, 2.84) | 0.489 |
| | Cholecystitis | 5893 | 664,749 | 1.47 (1.29, 1.68) | $4.71 \times 10^{-9}$* | 84.72 | 2.38 (1.57, 3.60) | $4.14 \times 10^{-5}$* |
| | Cholelithiasis | 42,510 | 664,749 | 1.20 (1.13, 1.27) | $5.75 \times 10^{-9}$* | 0.00 | 1.33 (1.02, 1.73) | 0.035 |
| | Acute pancreatitis | 6634 | 679,713 | 1.39 (1.23, 1.56) | $6.71 \times 10^{-8}$* | 79.71 | 1.55 (1.04, 2.31) | 0.031 |
| | Chronic pancreatitis | 3173 | 679,713 | 1.38 (1.17, 1.64) | $1.79 \times 10^{-4}$* | 0.00 | 1.27 (0.74, 2.16) | 0.384 |
| | Pancreatic cancer | 1643 | 701,472 | 1.00 (0.79, 1.26) | 0.999 | 67.21 | 2.08 (1.06, 4.10) | 0.034 |
| Other | Acute appendicitis | 25,361 | 690,149 | 1.15 (1.08, 1.23) | $1.27 \times 10^{-5}$* | 0.00 | 1.15 (0.92, 1.44) | 0.221 |

*Significant association after multiple testing.

UVMR = univariable Mendelian randomization. MVMR = multivariable Mendelian randomization. OR = odds ratio. CI = confidence interval. *Significant association after multiple testing.

**Table 2.** Associations of genetically predicted alcohol consumption with 24 gastrointestinal diseases in univariable and multivariable Mendelian randomization analyses.

| Disease | | Total cases | Total controls | UVMR OR (95% CI) | p Value | I² (95% CI) | MVMR adjusted for smoking initiation OR (95% CI) | p Value |
|---|---|---|---|---|---|---|---|---|
| Upper gastrointestinal diseases | Gastroesophageal reflux | 34,135 | 634,629 | 0.99 (0.81, 1.21) | 0.893 | 46.24 | 0.88 (0.72, 1.08) | 0.219 |
| | Esophageal cancer | 1130 | 702,116 | 2.86 (1.18, 6.91) | 0.020 | 22.68 | 1.28 (0.59, 2.82) | 0.533 |
| | Gastric ulcer | 8651 | 666,879 | 1.30 (0.95, 1.77) | 0.098 | 44.96 | 1.06 (0.77, 1.47) | 0.721 |
| | Duodenal ulcer | 5713 | 666,879 | 1.92 (1.23, 3.00) | 0.004* | 0.00 | 1.54 (1.01, 2.34) | 0.045 |
| | Acute gastritis | 3048 | 643,478 | 0.99 (0.58, 1.69) | 0.960 | 0.00 | 0.88 (0.52, 1.48) | 0.621 |
| | Chronic gastritis | 7975 | 643,478 | 1.33 (0.90, 1.95) | 0.147 | 77.04 | 1.33 (0.93, 1.89) | 0.115 |
| | Gastric cancer | 1608 | 701,472 | 1.57 (0.75, 3.30) | 0.233 | 0.00 | 1.59 (0.79, 3.21) | 0.194 |
| Lower gastrointestinal diseases | Irritable bowel disease | 15,718 | 641,489 | 0.73 (0.57, 0.93) | 0.012 | 11.84 | 0.74 (0.57, 0.97) | 0.027 |
| | Celiac disease | 4808 | 631,700 | 0.69 (0.44, 1.07) | 0.097 | 0.00 | 1.04 (0.64, 1.68) | 0.887 |
| | Diverticular disease | 50,065 | 587,969 | 0.95 (0.79, 1.13) | 0.553 | 67.29 | 0.94 (0.79, 1.13) | 0.527 |
| | Crohn's disease | 10,846 | 645,718 | 0.91 (0.62, 1.32) | 0.613 | 0.00 | 0.74 (0.53, 1.05) | 0.088 |
| | Ulcerative colitis | 16,770 | 651,255 | 1.11 (0.82, 1.50) | 0.509 | 0.00 | 0.88 (0.67, 1.15) | 0.358 |
| | Colorectal cancer | 9519 | 686,953 | 1.09 (0.76, 1.55) | 0.649 | 29.94 | 1.28 (0.95, 1.72) | 0.098 |

*Table 2 continued on next page*

*Table 2 continued*

| Disease | | Total cases | Total controls | UVMR OR (95% CI) | p Value | I² (95% CI) | MVMR adjusted for smoking initiation OR (95% CI) | p Value |
|---|---|---|---|---|---|---|---|---|
| Hepatobiliary and pancreatic diseases | Non-alcoholic fatty liver disease | 3242 | 707,631 | 1.20 (0.63, 2.28) | 0.574 | 0.00 | 0.99 (0.54, 1.79) | 0.962 |
| | Alcoholic liver disease | 2955 | 680,369 | 14.35 (7.69, 26.81) | $6.32 \times 10^{-17}$* | 92.68 | 9.60 (5.28, 17.46) | $1.25 \times 10^{-13}$* |
| | Cirrhosis | 5904 | 706,200 | 2.96 (1.50, 5.85) | 0.002* | 0.00 | 2.41 (1.29, 4.52) | 0.006* |
| | Liver cancer | 714 | 702,008 | 1.16 (0.43, 3.11) | 0.775 | 0.00 | 0.76 (0.29, 2.02) | 0.585 |
| | Cholangitis | 1708 | 664,749 | 0.96 (0.44, 2.08) | 0.912 | 0.00 | 0.72 (0.33, 1.55) | 0.397 |
| | Cholecystitis | 5893 | 664,749 | 1.36 (0.91, 2.03) | 0.132 | 84.72 | 0.96 (0.64, 1.45) | 0.862 |
| | Cholelithiasis | 42,510 | 664,749 | 1.02 (0.75, 1.39) | 0.878 | 0.00 | 1.03 (0.79, 1.35) | 0.801 |
| | Acute pancreatitis | 6634 | 679,713 | 1.36 (0.91, 2.03) | 0.128 | 79.71 | 1.17 (0.78, 1.75) | 0.456 |
| | Chronic pancreatitis | 3173 | 679,713 | 2.96 (1.80, 4.89) | $2.13 \times 10^{-5}$* | 0.00 | 3.24 (1.86, 5.64) | $3.18 \times 10^{-5}$** |
| | Pancreatic cancer | 1643 | 701,472 | 0.63 (0.32, 1.26) | 0.193 | 67.21 | 0.79 (0.40, 1.56) | 0.496 |
| Other | Acute appendicitis | 25,361 | 690,149 | 0.80 (0.63, 1.01) | 0.063 | 0.00 | 0.77 (0.61, 0.97) | 0.024 |

*Significant association after multiple testing.

UVMR = univariable Mendelian randomization. MVMR = multivariable Mendelian randomization. OR = odds ratio. CI = confidence interval.

for genetic liability to smoking initiation and multiple testing correction (*Table 2* and *Supplementary file 1H*).

## Discussion

We conducted a comprehensive MR investigation to examine the causal role of smoking and alcohol consumption in 24 gastrointestinal diseases, and the result summary of this comprehensive analysis is shown in *Figure 2* and *Supplementary file 1K*. We found robust associations between genetic predisposition to smoking and increased risk of 15 gastrointestinal outcomes independent of alcohol consumption, showing an extensive impact on gastrointestinal health. In contrast, genetically predicted alcohol consumption was robustly and predominantly associated with increased risk of liver and pancreatic diseases, including alcoholic liver disease, cirrhosis, and chronic pancreatitis after adjustment for smoking.

Corroborating and extending the previous observational studies, our MR investigation strengthened the evidence that smoking has a detrimental effect on gastrointestinal health and increases the risk of a broad range of gastrointestinal diseases, including gastroesophageal reflux disease (*Eusebi et al., 2018*), esophageal cancer (*Castro et al., 2018*), gastric and duodenal ulcer (*Kato et al., 1992*), gastritis (*Nordenstedt et al., 2013*), gastric cancer (*Zhang et al., 2020*), irritable bowel syndrome (*Talley et al., 2021*), diverticular disease (*Aune et al., 2017*), Crohn's disease (*Piovani et al., 2019*), cirrhosis (*Liu et al., 2009*), liver cancer (*McGee et al., 2019*), cholelithiasis (*Aune et al., 2016*), acute and chronic pancreatitis (*Aune et al., 2019*), and acute appendicitis (*Montgomery et al., 1999*). In line with previous MR studies, the current MR study also found that smoking was associated with increased risk of gastroesophageal reflux disease (*Yuan and Larsson, 2022a*), esophageal cancer (*Larsson et al., 2020*), gastric cancer (*Larsson et al., 2020*), diverticular disease (*Yuan and Larsson, 2022b*) non-alcoholic fatty liver disease (*Yuan et al., 2022c*), cholelithiasis (*Chen et al., 2022*), and acute and chronic pancreatitis (*Yuan et al., 2021*). As for ulcerative colitis, traditional observational studies revealed a decreased risk among current smokers (*Piovani et al., 2019*; *Park et al., 2019*); however, a recent MR analysis including 12,366 ulcerative colitis cases did not verify this inverse association in the analysis where smoking initiation was instrumented by 363 SNPs (*Georgiou et al., 2021*). Based on data from three independent populations, our study provided genetic evidence that smoking was a causal risk factor for ulcerative colitis in the analysis including 16,770 cases. Observational studies found that smoking was associated with an increased risk of colorectal cancer in a dose-dependent manner (*Botteri et al., 2020*), whereas the positive association was not observed in an MR analysis (*Larsson et al., 2020*). The current study was in line with the above MR study and found no strong association between smoking initiation and colorectal cancer risk. Nevertheless, a previous MR analysis with a 52,775 colorectal cancer cases found that genetic prediction to lifetime smoking index was positively associated with risks of colorectal cancer (*Dimou et al., 2021*), which might imply that our null finding might be caused by insufficient power due to a relatively small sample size. Smoking has been identified as a well-established risk factor for pancreatic cancer (*Mizrahi et al., 2020*). Interestingly, despite a null finding on the association of genetic liability to smoking initiation and pancreatic cancer in univariable MR analysis, the association became stronger and suggestively significant after adjusting for genetically predicted alcohol consumption. This might be explained by an inverse association between moderate alcohol consumption and pancreatic cancer. In addition, an adverse effect of smoking on pancreatic cancer was observed when using a smoking index as genetic instrument for lifetime smoking exposure. Our findings also provide novel evidence on the associations of smoking with the higher risk of cholecystitis and alcoholic liver disease independently of alcohol consumption, which need to be verified.

The pathogenic role of alcohol in alcoholic liver disease is well established and was confirmed also in our MR analysis. Our MR evidence along with previous observational studies also supported alcohol consumption as a risk factor for esophageal cancer (*Yu et al., 2018*), cirrhosis (*Roerecke et al., 2019*), and chronic pancreatitis (*Samokhvalov et al., 2015*). Noteworthy, the association between alcohol consumption and esophageal cancer became positively nonsignificant in multivariable MR, which possibly explained by the synergistic effect of alcohol and smoking. However, the association between alcohol consumption and duodenal ulcer has been scarcely studied. A meta-analysis including a small number of studies with relatively small sample sizes indicated that alcohol consumption was not associated with duodenal ulcer (*Ryan-Harshman and Aldoori, 2004*). This null finding

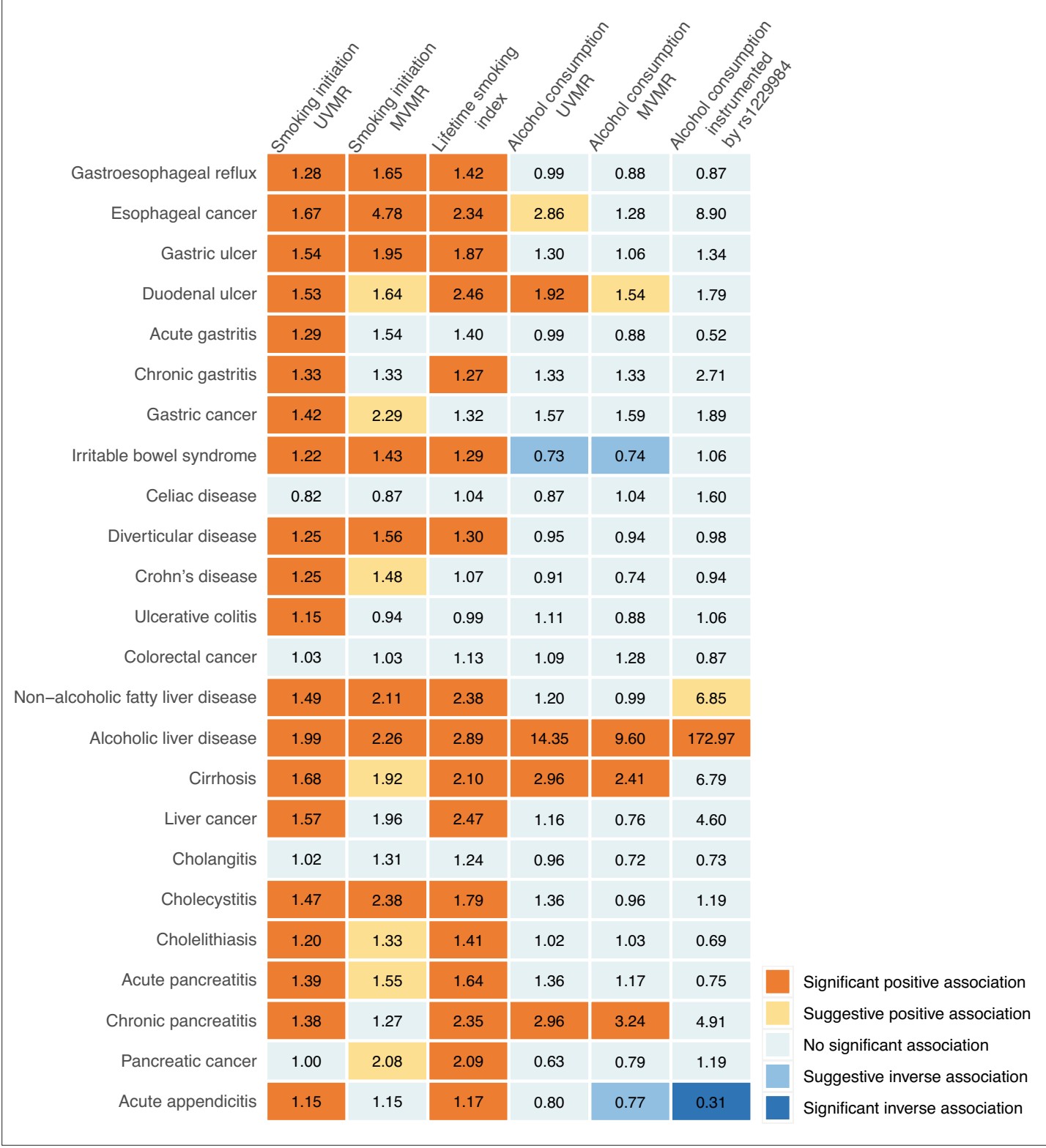

**Figure 2.** Summary of associations of genetically predicted smoking initiation, lifetime smoking, and alcohol consumption with 24 gastrointestinal diseases. UVMR, univariable Mendelian randomization; MVMR, multivariable Mendelian randomization. The numbers in the box are the odds ratios for associations of exposure for gastrointestinal diseases. The association with a p-value <0.05 but Benjamini-Hochberg adjusted p-value >0.05 was regarded suggestive, and the association with a Benjamini-Hochberg adjusted p-value <0.05 was deemed significant.

is likely due to insufficient power. Alcohol drinking has been associated with increased risk of gastric, colorectal, and liver cancer as well as acute pancreatitis (*Bagnardi et al., 2015*). These associations were not supported by our MR study. A possible explanation for this inconsistent findings is that heavy alcohol drinking is commonly associated with an unhealthy lifestyle and meager nutrition (*Klatsky, 2001*), which might exert confounding effects that could not be ruled out in previous observational studies. Another possible reason is that the U-shaped association could not be detected in MR analysis. For example, light drinking may be associated with decreased risk of these diseases (*McNabb et al., 2020*). In addition, it is also possible that the null associations observed in present study might be a consequence of inadequate power given SNPs used to mimic alcohol consumption explained a small phenotypic variance. In agreement with previous studies, our MR investigation demonstrated no associations of alcohol consumption with the development of gastroesophageal reflux, Crohn's disease, or ulcerative colitis (*Eusebi et al., 2018*; *Piovani et al., 2019*; *Georgiou et al., 2021*).

Many mechanisms have been proposed to support the observed positive associations between smoking and gastrointestinal diseases. Tobacco smoking has been shown to augment the production of numerous pro-inflammatory cytokines and decrease the levels of anti-inflammatory cytokines (*Arnson et al., 2010*), which might mediate a variety of inflammation-associated gastrointestinal diseases. In addition, smoking may also generate impacts on the immune system, including inhibition of the function of circulatory dendritic cells (*Givi et al., 2015*) and alteration signaling of Toll-like receptors (*Noakes et al., 2006*), which might contribute to the autoimmune disease and occurrence of neoplasm. The underlying mechanisms behind the associations of alcohol consumption with gastrointestinal diseases have not been fully understood. In addition to direct mucosal damage, the metabolites of ethanol are accountable for a part of the inflammation of alcohol drinking on the liver (*Mandrekar and Szabo, 2009*) and the gastrointestinal tract (*Bishehsari et al., 2017*).

This study investigated the impacts of smoking and alcohol consumption on a wide range of gastrointestinal disease. Based on our findings, promoting public awareness of the adverse impacts of tobacco smoking and alcohol consumption on gastrointestinal diseases is of particular importance and should be used as prevention strategies to lower gastrointestinal disease burden because these two factors are modifiable behavioral factors as possible targets of the pharmacal (*Leone et al., 2020*) and behavioral interventions. In addition, our results may help facilitate the guidelines of gastrointestinal disease prevention and the management of certain patients who have a subsequent high risk of gastrointestinal disease, like those with obesity and diabetes (*Camilleri et al., 2017*; *Krishnan et al., 2013*).

The major strength of the present study is MR design, which minimized bias from confounding and reverse causality and thus improved the causal inference in the associations of smoking and alcohol consumption with gastrointestinal diseases. We also used several independent outcome sources and combined the estimates, which increased statistical power as well as strengthened our findings by the observed consistency of results. Another strength is that we confined our analysis within the individuals of European ancestry, which minimized the population stratification bias.

This study also has several limitations. A major limitation of MR design is horizontal pleiotropy, which means that the used SNPs exert effects on the outcomes not via the exposure but via alternative pathways. However, in this study, the bias caused by pleiotropic effects should be minimal since we observed no indications of horizontal pleiotropy in MR-Egger analysis, consistent results from a series of sensitivity analyses, and robust associations from multivariable MR analysis with mutual adjustment. Another limitation is the relatively small phenotypic variance of alcohol consumption (approximately 0.2%), which resulted in inadequate power to detect weak associations for certain uncommon gastrointestinal diseases. There are several limitations of using summary-level data. First, we could not evaluate the nonlinear associations between alcohol consumption and gastrointestinal diseases without individual-level data. We could not differentiate the associations of smoking and alcohol consumption on the pathological subtypes of certain gastroenterological diseases, like esophageal cancer, based on summary-level data. For example, heavy alcohol consumption was associated with a high risk of squamous esophageal cancer (*Abnet et al., 2018*), but the associations were inconsistent for adenocarcinoma esophageal cancer (*Coleman et al., 2018*), which needs further investigation. Stratification analysis on sex was unlikely to be performed. In addition, we could not interpret and rescale the associations in a comparable scale to traditional observational studies because the unit of the exposure phenotypes was fixed in the corresponding genome-wide association analyses. An

additional limitation is that our analysis was confined to the European populations, and thus whether the observed associations can be generalized to other populations remains unknown. For alcohol consumption, it has been reported that there were substantial behavioral and genetic differences across ethnic groups. For example, East Asian individuals drink much less alcohol compared to other races, which appears to be related to *ALDH2* gene (*Jorgenson et al., 2017*). A further potential limitation is that the UK Biobank study was included in both the exposure and outcome datasets, which might cause MR estimates toward the observational associations. However, the used instrumental variants were proven to be strongly associated with the exposure (*F*-statistic >10) (*Burgess et al., 2016*), and the associations were replicated in the FinnGen study. Moreover, the associations remained stable in the sensitivity analyses using the genetic associations with exposures from the data excluding the UK Biobank and 23andMe studies. All of these indicated that the bias due to sample overlap was limited.

In conclusion, this MR study suggested that smoking is a risk factor for a broad range of gastrointestinal diseases independent of alcohol consumption. Alcohol consumption on the other hand seemed to be an independent risk factor for only a few gastrointestinal diseases, including alcoholic liver disease, cirrhosis, and chronic pancreatitis, but we cannot rule out weak associations with other diseases. These findings provide genetic evidence on supporting reducing tobacco smoking and possibly excessive alcohol consumption in particular to prevent gastrointestinal diseases.

## Acknowledgements

We want to thank the Lee Lab, the FinnGen study, the International Inflammatory Bowel Disease Genetics Consortium (IIBDGC), and the Genetic Epidemiology Research on Aging (GERA) for sharing data. XL: the Natural Science Fund for Distinguished Young Scholars of Zhejiang Province (LR22H260001). XYW: National Natural Science Foundation of China (81970494) and Key Project of Research and Development Plan of Hunan Province(2019SK2041); SCL: the Swedish Heart Lung Foundation (Hjärt-Lungfonden, 20210351), the Swedish Research Council (Vetenskapsrådet, 2019–00977), and the Swedish Cancer Society (Cancerfonden). Funders had no role in the design and conduct of the study; collection, management, analysis, and interpretation of the data; preparation, review, or approval of the manuscript; or the decision to submit the manuscript for publication.

## Additional information

### Funding

| Funder | Grant reference number | Author |
|---|---|---|
| National Natural Science Foundation of China | 81970494 | Xiaoyan Wang |
| Key Project of Research and Development Plan of Hunan Province | 2019SK2041 | Xiaoyan Wang |
| Hjärt-Lungfonden | 20210351 | Susanna C Larsson |
| Vetenskapsrådet | 2019-00977 | Susanna C Larsson |
| Cancerfonden | | Susanna C Larsson |
| Natural Science Fund for Distinguished Young Scholars of Zhejiang Province | LR22H260001 | Xue Li |

The funders had no role in study design, data collection and interpretation, or the decision to submit the work for publication.

### Author contributions

Shuai Yuan, Jie Chen, Conceptualization, Data curation, Formal analysis, Methodology, Writing - original draft; Xixian Ruan, Conceptualization, Methodology, Writing - original draft; Yuhao Sun, Dipender

Gill, Stephen Burgess, Edward Giovannucci, Conceptualization, Methodology, Writing – review and editing; Ke Zhang, Conceptualization, Writing – review and editing; Xiaoyan Wang, Conceptualization, Data curation, Funding acquisition, Writing – review and editing; Xue Li, Susanna C Larsson, Conceptualization, Data curation, Methodology, Writing – review and editing

## Author ORCIDs

Jie Chen ⓘ http://orcid.org/0000-0002-4029-4192
Xixian Ruan ⓘ http://orcid.org/0000-0002-4937-9168
Xiaoyan Wang ⓘ http://orcid.org/0000-0002-7281-1078
Xue Li ⓘ http://orcid.org/0000-0001-6880-2577
Stephen Burgess ⓘ http://orcid.org/0000-0001-5365-8760

## Ethics

Human subjects: Included studies had been approved by corresponding institutional review boards and ethical committees, and consent forms had been signed by all participants.

## Decision letter and Author response

Decision letter https://doi.org/10.7554/eLife.84051.sa1
Author response https://doi.org/10.7554/eLife.84051.sa2

## Additional files

### Supplementary files

• MDAR checklist

• Supplementary file 1. Supplementary material. (A) Information of included studies and consortia. (B) Definition of gastrointestinal diseases in UK Biobank and FinnGen. (C) Single nucleotide polymorphisms used as instrumental variables for smoking and alcohol consumption. (D) Power estimation of this Mendelian randomization analysis. (E) False discovery rate adjusted p values for all tested association. (F) Association of genetically-predicted smoke initiation with gastrointestinal disease in univaribale mendelian randomization. (G) Association of genetically-predicted smoking initiation (excluding 23andMe and UK Biobank), alcohol consumption (excluding 23andMe and UK Biobank) and lifetime smoking index with gastrointestinal disease in univariable mendelian randomization. (H) Association of genetically-predicted smoke initiation and alcohol consumption with gastrointestinal disease in multivaribale mendelian randomization. (I) Association of genetically-predicted alcohol consumption with gastrointestinal disease in univariable mendelian randomization. (J) Association of genetically-predicted alcohol consumption instrumented by rs1229984 in ADH1B with gastrointestinal diseases. (K) Evidence of causal association of smoking and drinking with 24 gastrointestinal diseases in the current study. (L) Association of genetically-predicted smoking initiation (excluding 23andMe and UK Biobank) with 24 gastrointestinal diseases. (M) Associations of genetically-predicted lifetime smoking with 24 gastrointestinal diseases. (N) Association of genetically-predicted alcohol consumption (excluding 23andMe and UK Biobank) with 24 gastrointestinal diseases.

• Reporting standard 1. STROBE-MR checklist for current Mendelian randomization study.

### Data availability

Data analyzed in the current study are publicly available GWAS summary-level data. The specific information and link could be found in Table S1, Supplementary file 1. The code and curated data for the current analysis are available at https://github.com/XixianRuan/smoking_gi (copy archived at swh:1:rev:1f31c18364102366be9ed05770e4a0f23de078f6).

*Continued on next page*

The following previously published datasets were used:

| Author(s) | Year | Dataset title | Dataset URL | Database and Identifier |
|---|---|---|---|---|
| Kurki MI | 2022 | FinnGen: Unique genetic insights from combining isolated population and national health register data | https://www.finngen.fi/en | The FinnGen study, finngen |
| Mengzhen L | 2019 | Data Related to Association studies of up to 1.2 million individuals yield new insights into the genetic etiology of tobacco and alcohol use | https://doi.org/10.13020/3b1n-ff32 | Data Repository for University of Minnesota (DRUM), 10.13020/3b1n-ff32 |

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
