## [Editor Report]

This is a valuable article that is methodologically convincing and provides evidence, through Mendelian Randomisation, that genetic predisposition to smoking and alcohol consumption influences the risk to develop different gastrointestinal diseases. The findings largely corroborate the findings from observational studies, especially for the effects of smoking. The major strength of the paper is the use of the largest possible genetic datasets for both the exposures and outcomes, which makes the findings more robust.

---

## [Decision Letter]

**Decision letter after peer review:**

Thank you for submitting your article "Smoking, Alcohol consumption, and 24 Gastrointestinal Diseases: Mendelian Randomization Analysis" for consideration by *eLife*. Your article has been reviewed by 2 peer reviewers, and the evaluation has been overseen by a Reviewing Editor and a Senior Editor. The following individual involved in the review of your submission has agreed to reveal their identity: Timothy Frayling (Reviewer #1).

As is customary in *eLife*, the reviewers have discussed their critiques with one another. What follows below is the Reviewing Editor's edited compilation of the essential and ancillary points provided by reviewers in their critiques and in their interaction post-review. Please submit a revised version that addresses these concerns directly. Although we expect that you will address these comments in your response letter, we also need to see the corresponding revision clearly marked in the text of the manuscript. Some of the reviewers' comments may seem to be simple queries or challenges that do not prompt revisions to the text. Please keep in mind, however, that readers may have the same perspective as the reviewers. Therefore, it is essential that you attempt to amend or expand the text to clarify the narrative accordingly.

Essential revisions:

Your manuscript has been positively reviewed by two independent Reviewers. Based on their comments and the discussion afterwards, we came up with the following main point that needs to be addressed in addition to specific ones brought up by the reviewers individually (see reviews below).

– Please provide a more thorough assessment of the genetic variants to evaluate the potential for pleiotropy between the genetics of both tobacco and alcohol and other risk factors. One analysis that can be done to address this would be Steiger filtering.

*Reviewer #1 (Recommendations for the authors):*

This is a thorough and powerful Mendelian Randomisation analysis that tests the role of two very important risk factors for disease – smoking and alcohol consumption – as risk factors for a range of key gastrointestinal conditions. It is important as the approach greatly reduces the influence of confounding and reverse causation and provides insight into the extent to which these behaviours (genetically influenced ) lead to disease.

The results are refreshingly very clearly presented as univariable and multi-variable forest plots (Figure 2).

1. In a similar way to the use of a single main snp with a known mechanism for alcohol consumption (ADH1B) what were the smoking results when using the chromosome 15 variant(s)? The Wald test results from these

2. Meta-analysis of outcomes. Two or three sources of genetic data were used for the GI outcomes. It is not completely clear how they were meta-analysed – at the individual SNP level or the MR result level. From the stats section, I think it was at the individual SNP level but was there any evidence of heterogeneity of individual snps? I think it important to present results at the MR level as well to see if the causal effect estimates are consistent across ukbb, finger, and the disease-specific GWAS where relevant.

3. In the Results section how do we interpret the odds ratios? I suggest useful to remind readers that these are per unit alcohol, N cigarettes, etc.

4. Can the authors give the reader a quick sense of how many GWAS signals for smoking and alcohol overlap? Even if not exactly the same snp I imagine many index signals for risk-taking behaviour for example overlap.

*Reviewer #2 (Recommendations for the authors):*

The authors have brought together extensive amounts of genetic data in order to conduct Mendelian randomization analyses for 24 gastrointestinal diseases. The immediate goal is to remove the possibility of confounding and reverse causation from these associations for alcohol and tobacco, and enable causal associations to be inferred. The secondary goal is to enable further recommendations for public policies and clinical interventions.

As very few readers are likely to be experts in the evidence of alcohol and tobacco and these 24 diseases, a clear presentation of where the current doubt lies would be helpful. Decades of observational-based studies have been published on all of these diseases, and many rigorous evaluations have been conducted on levels of causality (eg including WCRF and Monographs for specific cancers). Is reverse causality (ie individuals smoking more or drinking more based on early symptoms) really a hypothesis for any of these diseases? If so, it would be useful to outline them. Confounding is always difficult to rule out, although the extensive evidence on dose response, quitting, etc has led to conclusions of causality for many of these diseases. Confounding (pleiotropy) with genetic studies is also a major concern. A comprehensive evaluation of the potential for pleiotropy between the genetics of both tobacco and alcohol and other risk factors would appear necessary.

A supplementary table indicating for which alcohol and tobacco associations the evidence for causality was in doubt would be helpful.

---

## [Author Response]

Reviewer #1 (Recommendations for the authors):This is a thorough and powerful Mendelian Randomisation analysis that tests the role of two very important risk factors for disease – smoking and alcohol consumption – as risk factors for a range of key gastrointestinal conditions. It is important as the approach greatly reduces the influence of confounding and reverse causation and provides insight into the extent to which these behaviours (genetically influenced ) lead to disease.The results are refreshingly very clearly presented as univariable and multi-variable forest plots (Figure 2).

We thank the Reviewer for reviewing our paper and providing positive feedbacks as well as constructive comments. We have now carefully considered these comments and revised the paper accordingly.

1. In a similar way to the use of a single main snp with a known mechanism for alcohol consumption (ADH1B) what were the smoking results when using the chromosome 15 variant(s)? The Wald test results from these

Many thanks for this comment. We agree with the Reviewer that using a single SNP in chromosome 15, like rs1051730, to genetically mimic smoking behaviour is a good way to minimize horizontal pleiotropy. However, the analysis based on a single SNP that explains limited phenotypic variance had very low power, and the results of this analysis with wide confidence intervals were unreliable. In addition, smoking is a complex phenotype that may not be well proxied by a single SNP. Thus, we did not include this analysis in the study.

2. Meta-analysis of outcomes. Two or three sources of genetic data were used for the GI outcomes. It is not completely clear how they were meta-analysed – at the individual SNP level or the MR result level. From the stats section, I think it was at the individual SNP level but was there any evidence of heterogeneity of individual snps? I think it important to present results at the MR level as well to see if the causal effect estimates are consistent across ukbb, finger, and the disease-specific GWAS where relevant.

Many thanks for this comment. Given that we aimed to examine the consistency of the associations between studies, the meta-analyses were conducted at MR level instead of SNP level, which is consistent with the approach recommended by the Reviewer. For each association, the heterogeneity of SNPs’ estimates was evaluated by Cochran's Q test, the result of which is presented in the supplementary tables. Regarding the heterogeneity of the associations between studies, we have now estimated the I^2^ statistic. Overall, low heterogeneity was observed for most associations. We have added I^2^ statistic in Figure 2 and 3. In addition, we observed similar point estimates and overlapped confidence intervals of the associations between studies, which also indicated low heterogeneity of the associations. We apologized for the unclear description, and we revised the corresponding sentences as follow:

Page 9:

“Estimates for each association from different sources were combined using fixed-effects meta-analysis and the heterogeneity of the associations from different data sources were evaluated by the I^2^ statistic. Heterogeneity among SNPs’ estimates in each association was assessed by Cochran’s Q value.”

3. In the Results section how do we interpret the odds ratios? I suggest useful to remind readers that these are per unit alcohol, N cigarettes, etc.

Many thanks for this comment. It is difficult to interpret the odds ratio in MR studies based on summary-level data where the unit of the exposures were fixed. In this study, we unfortunately could not interpret or rescale the odds ratio in a direct way as expected by the Reviewer due to lack of individual-level data, which also confined the comparison of our results with previous observational findings with respect to the magnitude of the associations. However, our findings are still of clinical and public health significance since the MR study strengthened the causality of the associations, which emphasized the roles of smoking in particular and alcohol consumption in the development of gastroenterological diseases. We have now discussed this in the manuscript.

Page 19:

“In addition, we could not interpret and rescale the associations in a comparable scale to traditional observational studies because the unit of the exposure phenotypes were fixed in the corresponding genome-wide association analyses.”

4. Can the authors give the reader a quick sense of how many GWAS signals for smoking and alcohol overlap? Even if not exactly the same snp I imagine many index signals for risk-taking behaviour for example overlap.

Many thanks for this comment. There were just two index signals shared by smoking initiation and alcohol consumption, which are rs1713676 and rs11692435. We are not able to assess this for lifetime smoking index due to lack of summary-level data. We have now added this information in the manuscript.

Page 6:

“Smoking initiation and alcohol consumption shared two index genetic variants, which were rs1713676 and rs11692435.”

Reviewer #2 (Recommendations for the authors):The authors have brought together extensive amounts of genetic data in order to conduct Mendelian randomization analyses for 24 gastrointestinal diseases. The immediate goal is to remove the possibility of confounding and reverse causation from these associations for alcohol and tobacco, and enable causal associations to be inferred. The secondary goal is to enable further recommendations for public policies and clinical interventions.

We sincerely thank the Reviewer for reviewing our paper and providing insightful comments. We have now carefully considered these comments and improved the manuscript accordingly.

As very few readers are likely to be experts in the evidence of alcohol and tobacco and these 24 diseases, a clear presentation of where the current doubt lies would be helpful.

Many thanks for this comment. Previous epidemiological studies provided quite consistent evidence in support of positive associations of smoking and alcohol consumption with increased risk of many common gastroenterological diseases. The motivation of this study was (1) to strengthen the causality of these associations using MR approach; and (2) to explore the associations of smoking and alcohol consumption with scarcely studied gastroenterological diseases. We have briefly introduced the background as well as the motivations of the study in the Introduction part.

Page 3:

“Population-based studies have identified tobacco smoking as a risk factor for several gastrointestinal diseases, including gastroesophageal reflux disease (4), esophageal cancer (5), Crohn’s disease (6), liver cancer (7), and pancreatitis (8). Evidence on the association between tobacco smoking and risk of other gastrointestinal diseases is limited and inconsistent. Like smoking, heavy alcohol consumption has been associated with increased risk of several gastrointestinal outcomes, including gastritis (9), gastric cancer (10), colorectal cancer (11), cirrhosis (12), liver cancer (7), and pancreatitis (8). However, whether these associations are all causal remain unestablished since most of the evidence was obtained from observational studies where the results may be biased by reverse causality and confounding. As smoking and alcohol consumption are phenotypically and genetically correlated (13, 14), the independent impacts of smoking and alcohol consumption on gastrointestinal diseases are unclear. Establishing the causal association of tobacco smoking and alcohol consumption with gastrointestinal diseases is crucial, as this provides further evidence for subsequent recommending public health policies and clinical interventions.”

Decades of observational-based studies have been published on all of these diseases, and many rigorous evaluations have been conducted on levels of causality (eg including WCRF and Monographs for specific cancers). Is reverse causality (ie individuals smoking more or drinking more based on early symptoms) really a hypothesis for any of these diseases? If so, it would be useful to outline them. Confounding is always difficult to rule out, although the extensive evidence on dose response, quitting, etc has led to conclusions of causality for many of these diseases. Confounding (pleiotropy) with genetic studies is also a major concern. A comprehensive evaluation of the potential for pleiotropy between the genetics of both tobacco and alcohol and other risk factors would appear necessary.

Many thanks for the comment. Reverse causality should not be a common issue for any of studied gastroenterological diseases. However, this bias might exist for certain gastroenterological diseases causing pain, which smoker patients may try to increase smoking dose to mitigate via an intake of higher levels of nicotine. We have now discussed this in the manuscript. We agree with the Reviewer that pleiotropy is a major limitation of any MR studies. In the current study, we have taken care of possible pleiotropic effects by examining through MR-Egger and MR-PRESSO analyses. In these analyses, we found limited evidence in support of horizontal pleiotropy as a major issue biasing our findings. The strongest pleiotropic effects may be caused by a genetic correlation between smoking and alcohol consumption. Thus, we conducted multivariable MR analysis to adjust for genetically predicted alcohol consumption in the analysis of smoking and vice versa. The consistency of the associations in these analyses indicated that this pleiotropy should not be a major issue. We did not perform multivariable MR with adjustment for other factors due to potential collider bias that can be introduced by additional adjustment in MR analysis.

Page 3:

“Of note, even though reverse causality may not be an issue in the studies for any of studied gastroenterological outcomes, it might exist for certain gastroenterological diseases causing pain, which smoker patients may try to increase smoking dose to mitigate via an intake of higher levels of nicotine.”

A supplementary table indicating for which alcohol and tobacco associations the evidence for causality was in doubt would be helpful.

Many thanks for this comment. We have summarized the evidence of the current analysis in S11 Table to indicate what associations was certain or in doubt.